# Learning a Path from Real Navigation: The Advantage of Initial View, Cardinal North and Visuo-Spatial Ability

**DOI:** 10.3390/brainsci10040204

**Published:** 2020-04-01

**Authors:** Veronica Muffato, Chiara Meneghetti

**Affiliations:** Department of General Psychology, University of Padova, 35131 Padova, Italy

**Keywords:** navigation, allocentric frame of reference, egocentric frame of reference, cardinal points, initial headings, visuo-spatial ability, sense of direction

## Abstract

Background: Spatial cognition research strives to maximize conditions favoring environment representation. This study examined how initial (egocentric) navigation headings interact with allocentric references in terms of world-based information (such as cardinal points) in forming environment representations. The role of individual visuo-spatial factors was also examined. Method: Ninety-one undergraduates took an unfamiliar path in two learning conditions, 46 walked from cardinal south to north (SN learning), and 45 walked from cardinal north to south (NS learning). Path recall was tested with SN and NS pointing tasks. Perspective-taking ability and self-reported sense of direction were also assessed. Results: Linear models showed a better performance for SN learning and SN pointing than for NS learning and NS pointing. The learning condition x pointing interaction proved SN pointing more accurate than NS pointing after SN learning, while SN and NS pointing accuracy was similar after NS learning. Perspective-taking ability supported pointing accuracy. Conclusions: These results indicate that initial heading aligned with cardinal north prompt a north-oriented representation. No clear orientation of the representation emerges when the initial heading is aligned with cardinal south. Environment representations are supported by individual perspective-taking ability. These findings offer new insight on the environmental and individual factors facilitating environment representations acquired from navigation.

## 1. Introduction

Navigation is an essential everyday activity. Maximizing the conditions that support this ability is one of the priorities of spatial cognition research. The conditions that favor (or hinder) navigation include how information is acquired from one’s own point of view, depending on the position of one’s body in a space, and the features of an environment, given by the area as a whole and how it is located in reference to the world-based frame, in terms of cardinal directions. These conditions may relate to the individuals themselves, with an important part played by their abilities and personal preferences and approach to the environment [1,2]. Conjugating the analysis of several aspects offers interesting insight on the conditions favoring environmental knowledge acquisition by navigation. The aim of the present study was to examine the learning conditions that can favor navigation by considering the view from the individual’s body associated with world-based information provided by the environment. Specifically, we aimed to shed new light on the effect of the cardinal points in influencing how information is organized in people’s mental representations and how it relates to their personal abilities and preferences.

### 1.1. Navigation: Egocentric and Allocentric (World -Based) Frames of Reference

Navigation is a complex process by means of which a space is experienced from an egocentric point of view, based on sensorimotor information about an individual’s position in space, self-to-object distances, and self-motion. This information enables individuals to learn a sequence of landmarks, turns, and changes of direction, and to memorize a set of place-action associations [2]. Learning spatial information by navigating in an environment gives rise to a mental representation, or cognitive map [3], that can be expressed in various ways, such as retracing the same route or finding new routes after exploring a new environment (environment-based tasks), or estimating distances and directions (spatial information managing tasks). The distinction between the egocentric and the allocentric frames of reference is a concept fundamental to our understanding of how spatial information is encoded and organized in our memory (e.g., [4,5]). The egocentric frame considers spatial information in relation to our own body (self-to-object relations) while the allocentric frame envisages elements (landmarks) in the environment in relation to one another (object-to-object relations). These two frames can also be described respectively as a Polar reference system (where the pole is the body, on which the other relations are established) and a Cartesian reference system (where points in space are given by global coordinates). When environmental information is acquired by navigation, the person’s view becomes the main frame of reference for encoding and memorizing information. This is a clear indication of the use of the egocentric frame. When landmarks are mentally represented in relation to one another, and to an overall environment, as elements in a layout (such as buildings passed along a path travelled within the perimeter of an area), this is an indication of the use of the allocentric frame of reference [6,7].

In humans, both egocentric and allocentric frames are involved in delineating the final features of spatial representations [4]. It has been demonstrated that animals, from the simpler species (like rats and bats) to the most advanced (such as chimpanzees), can also orient themselves using egocentric and allocentric information [8,9,10]. Studies in this domain use different sources for the acquisition of spatial information, based on the perception of actual elements arranged in a configuration (e.g., [11]), or from map learning (e.g., [12]), for instance. Then, one of the ways often used to assess the properties of environment representations is to use a pointing task (also termed the judgment of relative direction [11]). This task involves using spatial memory to judge relative directions of landmarks after adopting different imaginary views. An individual is asked to imagine standing at a given landmark (or object) in an environment (or configuration), facing another, and pointing in the direction of a third (target) landmark. People are generally more accurate in pointing at landmarks aligned with their initial view (the one they had adopted during the learning phase) than in pointing from misaligned (or counter-aligned) views, when they have to adopt a different imaginary view from the one used to encode the information [13]. The facilitation deriving from the actual initial view adopted during the learning phase being aligned with the imagined view during recall is called the “alignment effect” [14,15]. The learners’ initial view, based on experience, anchors the way the information being memorized is organized, serving as a principal reference vector that helps them to mentally represent an environment’s structure. This egocentric exposure establishes the orientation of their whole mental representation, which acquires the properties of a “conceptual north” (a personal north-up orientation). At the same time, a part of the environmental information, such as the wall of a room [11], or the whole shape of a layout [12], can be integrated in the learner’s mental representation. Overall, studies on static spatial learning based on looking at elements in a layout [11,12] show that the initial view experienced (egocentric frame) has an essential role, but it is combined with environmental information (allocentric frame) to obtain an integrated mental representation.

When we specifically consider learning from navigation, be it by exploring or moving along a preset path in an actual or virtual environments, the orientation of a spatial representation is based mainly on the learner’s initial view [7]. There is also evidence of the view adopted in the first direction of travel (such as the first leg of a route) being particularly relevant in deciding the orientation of the representation as a whole. This is called the “first perspective alignment effect”. In navigation, the effect of the initial heading view can be attenuated by other information, which is not egocentrically based. When navigating irregular spaces, such as when we chart a path that goes in different directions, we incorporate multiple local views in our representations [12,16]. We include environmental information when an irregular path is charted through a visible layout of the space (such as the walls of a room [17]), or when the shape of the path we cover is dissonant with central landmarks [18], or when we move along a path with landmarks visible some distance away [19]; see also, [20]. In short, while we are navigating in an environment, as we encode spatial information from our own point of view, the representation we form in our memory contains mainly egocentric information, especially when the path we cover is regular and consistent with the environment’s landmarks and layout. This representation can be integrated with allocentric information, however, when paths are more complex and irregular and features of the environment are visible and salient.

As we navigate in the environment, our allocentric frame of reference can be enlarged, in the absolute sense, using world-based information, such as when cardinal points are taken for reference in an environment representation. Cardinal points can influence the representation’s properties. There is evidence to show that representations of familiar environments (such as a person’s home town or university campus) are aligned with the cardinal north [21,22]. For instance, Frankenstein and colleagues [21] asked a sample of people to face in various directions ranging from 0° (cardinal north) to 180° (south) in their home town, and to point towards familiar landmarks. People made more mistakes, the more they deviated from a cardinal north-facing orientation. Marchette and colleagues [22] asked their participants to point at buildings on a university campus from imagined body positions, and found a cardinal north-up advantage even when local maps were typically oriented westwards. These studies suggest that our environment representations rely on a cardinal north-up orientation, for familiar environments at least. On the other hand, other studies on familiar environments found cardinal north less important in orienting representations [23,24] than specific features of an environment (such as main roads, buildings, or natural landmarks) [25]. It should be noted, however, that the latter studies considered familiar environments in which multiple exposures from different source would be experienced. The role of cardinal points in forming environment representations after learning from navigating new environments has yet to be fully explored, but they could be expected to play a part in the way spatial information is organized in our memory.

There is a need to gain a better understanding of the interplay between egocentric and allocentric (world-based) frames of reference in forming spatial representations when a new environment is learnt using such a common modality as navigation. This would enable us to examine to what extent information acquired by navigating (from a primarily egocentric experience) can incorporate world-based information, such as cardinal points. The few studies available on environment representations of newly acquired spatial information seem to support a central role for initial heading orientation [26,27], in which the influence of the allocentric reference frame is detectable [28]. In a study by Iachini and Logie [26], participants unfamiliar with a campus were first shown a target building, and then led to see another view of the same building (at a 0–180° angle from their starting position). Then they had to locate their own actual position on a three-dimensional map. The results showed faster response times when the initial view and the test position were aligned, compared with other test positions that moved away from the initial view. Brunyé and colleagues [27] studied route planning in unfamiliar real environments with a route selection task. They found a strong influence of initial segment straightness and topography of the environment (mountain, level of terrain), and a relatively weak influence of cardinal directions. These results seem to support the core role of initial learning view. Gagnon and colleagues [28] assessed the properties of mental representations in relation to allocentric (world-based) reference points derived from navigation. At the entrance to a virtual city, participants were shown an initial allocentric heading with a compass rose pointing north, south, east or west, that disappeared before they entered the environment. Participants first navigated freely in the environment (and the total length of the path they took and the number of stops were recorded), then they completed a spatial statements verification test. The results showed that participants exposed to a northward initial orientation traveled longer distances, made fewer stops, and were faster at assessing spatial relations between landmarks than when they were initially exposed to a southward, eastward or westward orientation. This suggests that environment representations derived from navigation, and their flexibility, can be influenced by exposure to a declared northward orientation presented with the initial heading. It is important to note that, in their study, Gagnon and colleagues established the cardinal orientation by displaying a compass rose at the beginning of the navigation phase; this cardinal orientation was not that of the real world. They did not use a pointing task as an informative measure to assess the orientation of the representation, as done in other studies [e.g., 6,22]. These aspects (not considered by Gagnon and colleagues) offer insight on the importance of examining the orientation properties of a representation considering the real cardinal north (as a learning modality) and pointing (as an assessment modality), as done in the present study.

Other studies used route descriptions, i.e., describing an imaginary path through an environment from a person’s point of view, in which the direction of the cardinal north was explicitly indicated [28,29,30,31]. They offer some interesting insight, revealing similar orientation effects with visual and verbal input (e.g., [19]). These studies indicate that it was easier to form spatial representations with a northward orientation when it was explicitly mentioned that an imaginary path started heading northwards, rather than when it was described as heading southwards. In particular, the results showed that participants were more accurate in northward than in southward pointing task [29,30] (see also [31]), they processed spatial information more quickly and formed mental representations that enabled them to adopt different points of view [28]. When a mismatch was created between a declared initial north-oriented view and the orientation of the remainder of an imaginary path (presenting the information mainly from an east-west orientation), the beneficial effect of the initial view disappeared [30].

These findings suggest that the initial view of a path (be it virtually navigated or imagined from verbal descriptions) decides the orientation of the whole representation [19,20]. When the initial view of an environment being learned (egocentric; personal north-up) is congruent with the cardinal north (i.e., world-based information), the resulting environment representation is clearly north-oriented and this seems to facilitate its accuracy [28,29,30]. In other words, when we navigate in an environment, egocentric information (gained from the initial heading) is essential for the purposes of anchoring the orientation of the environment in our minds. At the same time, our mental representation is better defined when it is congruent with world-based information given by compass north. Given this evidence, we still need to know more about how egocentric information (initial heading) acquired from navigation can be integrated with allocentric (world-based) information. The main aim of the present study was therefore to examine whether the features of environment representations formed after navigation follow a particular orientation due mainly to the initial egocentric view, or to a combination of this initial egocentric view with allocentric (world-based) information.

### 1.2. Navigation: The Role of Individual Factors

Several studies found navigation learning accuracy related to individual visuo-spatial factors [29,32,33,34,35,36]. These factors include both objectively measurable visuo-spatial abilities, such as mental rotation (as measured by the Mental Rotations Test [37]) and perspective taking (as measured with the Object Perspective Task (OPT; [38]), and self-reported wayfinding competence and preferences, such as sense of direction (i.e., an individual’s estimation of their own ability to orient themselves in the environment [39,40]). It has been demonstrated that individuals’ ability to adopt different imaginary views in a perception-based object configuration (as in OPT) can help to predict their performance in tasks that involve adopting memory-based imaginary views of an environment learnt by navigation [29,30,41,42,43]. In fact, higher OPT scores predict greater accuracy in pointing in the direction of landmarks in an environment (i.e., fewer degrees of error), and there is evidence of this involvement being more obvious when pointing in the test phase is misaligned with the initial view of the environment learning phase [29,30,41]. Being convinced of having a good sense of direction also correlates with a greater ability to adopt different orientations in an environment [27,44,45]. It should be noted that when visuo-spatial abilities and self-reported wayfinding attitudes (as sense of direction) are considered as distinct factors, the former affect performance to a greater degree than the latter [32,35]. The role of individual visuo-spatial factors in learning from navigation in relation to egocentric and allocentric (world-based) frames of reference deserves to be better examined. A second aim of the present study, related to the first, was therefore to examine the role of visuo-spatial abilities (especially perspective taking), and self-reported sense of direction in supporting environment representations (which is potentially based on the interplay of egocentric and allocentric information).

### 1.3. Rationale and Aim of the Study

Based on the above review of the literature, this study aimed to consider the case of learning a new environment by navigation, examining: (i) the interplay of egocentric and allocentric (world-based) frames of reference (i.e., whether the initial egocentric view prevails, or the integrated egocentric and allocentric information) in forming a mental representation of the new environment and organizing it in memory; and (ii) to what extent this representation is supported by individual visuo-spatial factors. To accomplish these aims, we chose an unfamiliar natural setting—a park—with a path mainly oriented in the direction of the cardinal north or south. This setting allows us to distinguish the effect of the initial view from that of the world-based information (cardinal north- or south-oriented) in participants’ environment representations. An outdoor environment was chosen in the light of the evidence of the influence of the cardinal coordinate system in the experience of outdoor environments [46,47,48]. A group of undergraduates walked along the path, half of them moving from the south to the north side of the park (SN-learning condition), and the other half in the opposite direction, north to south (NS-learning condition). A pointing task was administered, asking participants to imagine facing south or north and pointing in other directions. This type of task enables the specific orientation of an individual’s mental representation of an environment to be assessed. A map-drawing task, which involved locating landmarks on a sketch map, was used as a control measure of the clarity of participants’ mental representations of the landmarks in the environment. Individual visuo-spatial abilities (i.e., the OPT) and self-reported sense of direction were assessed.

The following hypotheses could be envisaged.

(i) Path learning: egocentric and allocentric (world-based) frames of reference

Two possible results might be expected:

(a) the egocentric experience may prevail in the organization of the mental representation, given by the initial heading [e.g., 7,19] for our regular path through an environment with regular features and no dissonant elements (whereas a complex path with dissonant environmental elements can reduce the initial heading effect [17,18,19]). This being the case, pointing performance should be better when aligned with the initial heading experienced because it would define the conceptual north that anchors the orientation of the whole representation. The group moving from south side (SN group) should therefore perform better in SN pointing than in NS pointing and, vice versa, the group starting on the north side (NS group) should better in NS pointing than in SN pointing;

(b) the organization of the mental representation may integrate both the initial view (egocentric experience) and allocentric (world-based) information (as suggested by studies using virtual navigation [28], and verbally-presented paths [29,30,31], with a preference for a cardinal northward orientation [25,26]). If this is the case, the representation may be organized according to a specific northward orientation when the initial heading is oriented towards compass north, as in the group taking the path from south to north (SN group), who would perform better in SN pointing than NS pointing. On the other hand, for the group walking from north to south (NS group), the initial view (personal north-up) is aligned not with the cardinal north, but with the cardinal south, and this discordance might disrupt the effect of the initial view [28,29], so that there would be no noteworthy difference in this group’s performance in two types of pointing.

(ii) Path learning and individual visuo-spatial factors

The individual’s perspective-taking ability should support their mental representations of an environment they have navigated [41,42,43]. It may be that this ability is more heavily involved in the test phase in generating imaginary views counter-aligned with that of the learning phase [29,30]. An individual’s estimation of their personal sense of direction may also influence their pointing accuracy, given its involvement in the orientation of their mental representations of an environment [27,44].

## 2. Materials and Methods

### 2.1. Participants

Initially, a group of 140 undergraduates at Padova University’s School of Psychology completed a questionnaire on their familiarity with the city’s “Europe park”, not far from the buildings where their lectures were held (the number of times they had gone there and their reasons for doing so). From this initial group, a sample of 91 respondents (76 females, *M* age = 21.89, *SD* = 2.25) reported knowing nothing about the park and never having been there. These respondents were randomly assigned to two path learning conditions: 46 walked the path from south to north (SN-learning condition), the other 45 from north to south (NS-learning condition). Based on power analyses run with the “pwr” library in R for linear models, it was calculated that 36 participants were needed to obtain a power of 0.80 and an effect size of 0.30. The study was approved by the Ethics Committee for Research in Psychology (University of Padova; No. 2744) and complied with the fundamental principles established in the Declaration of Helsinki.

### 2.2. Materials

#### 2.2.1. Learning Phase: The Path

The path considered is a cycle and footpath through the Europe Park in Padova, an area about 160 × 300 m (4.8 ha surface area) surrounded by a wall. The path is around 330 m long and passes roughly through the middle of the longest part of the park, a condition in which there is no dissonance between the path and environmental information (i.e., the shape of the layout). This was a deliberate choice because any such dissonance has an effect on a person’s mental representations (e.g., [17,18]). The park has four gates (one for each cardinal point, north, south, east and west) and a number of interconnected paths (see Figure 1). The points of departure or arrival of our chosen path are Gate 1 (cardinal north oriented) and Gate 2 (cardinal south oriented). Eight landmarks are encountered along the path (see Figure 1); a previous pilot study had ascertained that they were readily visible from the path and easy to remember.

Question to test knowledge of cardinal north. Participants standing at their starting point (in front of the gate) were asked: “Where is the north from here?”, and they used their own arm to give their answer, which was marked in a circle in a sheet of paper by the experimenter. Their answers were considered as correct when they came within 30° of the right direction (score: 1 for correct answers; 0 otherwise).

#### 2.2.2. Recall Phase

Pointing task. A computerized version of the pointing task was implemented using E-Prime 1; Psychology Software Tools; Sharpsburg, USA) and projected onto a screen (32-inch Ultra HD 4K 3840 × 2160 LED monitor; LG Electronics Italia; Milan, Italy). A sentence written in white on a black screen instructed participants to imagine standing at a given landmark, facing another and pointing in the direction of a third. A circle was simultaneously displayed in the middle of the screen. An arrow pointing upwards from the center of the circle indicated the participant’s position (at the center) and which way they were facing. The arrow could be moved around the perimeter (using a mouse) to indicate the direction of the third landmark. The task consisted of two familiarization items and 24 test items, 12 that involved facing north (e.g., “Imagine standing on the hill and facing the seal statue, then point to the rushes”; SN pointing), and 12 that involved facing south (e.g., “Imagine standing at the listening point and facing the glass house, then point to the seal statue”; NS pointing). The score corresponded to the lesser angle between a participant’s answer and the correct direction for each item, and the mean of their degrees of error was computed for the SN and NS pointing items.

Map drawing. Two sketch maps of the park were prepared, one for each learning condition, showing the perimeter of the area and an arrow indicating the starting points, which headed north or south. Participants were asked to locate landmarks in their right positions. Their global accuracy was calculated using the Square Root of the Canonical Organization (SQRTCO; score: 0-1) index in the GMDA program (Gardony Map Drawing Analyzer) [49], comparing each landmark’s position relative to all the other landmarks using NSEW (North, South, East and West) directions.

#### 2.2.3. Visuo-Spatial Measures

Short Object Perspective Test (sOPT [50]; adapted from [38]). This task involves having to imagine standing at one object in a layout of 7 objects, facing another, and pointing towards a third (misaligned by at least 90° vis-à-vis what respondents imagine they are looking at). Participants indicate the direction by drawing an arrow from the center of a circle to its perimeter. The task consists of 6 items and has a time limit of 5 min. The score can range from 0° to 180° mean degrees of error (Cronbach’α = 0.62 in the present sample).

Sense of Direction and Spatial Representation questionnaire (SDSR [51]). This questionnaire assesses respondents’ convictions regarding their sense of direction and wayfinding preferences. It is composed of 13 items covering self-reported sense of direction (e.g., “Do you think you have a good sense of direction?”), a preference for a map-based or route- or landmark-based wayfinding mode, and knowledge and usage of cardinal points. Answers are given using a Likert scale from 1 (not at all) to 5 (very much). A total score is calculated from the sum of all the ratings and considered as an overall measure of self-reported environment representation ability [52]; max: 65; Cronbach’α = 0.80 in the present sample).

### 2.3. Procedure

Participants signed the informed consent explaining the aim and the procedure of the study. Then they attended two individual sessions, one in the field (30 min), followed by a second at the lab in the psychology department (30 min), plus the 10–15 min walk from the psychology department to the park and back again. The field session was always conducted early in the morning (from 8 to 10 am) to ensure that the learning conditions were the same for all participants and to avoid the park being crowded. The experimenter met each participant outside the psychology department building and walked with them to Gate 1 (south entrance) or Gate 2 (north entrance) about 400 m away, depending on the learning condition randomly assigned to them (ensuring that males and females were assigned to both learning conditions). On arrival at the entrance to the park, the experimenter asked the participant to indicate the direction of the cardinal north with their arm and recorded their answer (without giving any feedback). This was done because of its impact on knowledge acquisition [46,47]. Then participants were explicitly told that they were at the north or south gate (in the NS- and SN-learning conditions, respectively). The experimenter instructed them to memorize the path they were about to walk along, and all the elements that would be named and pointed at along the way. The experimenter specified that they would need this information in further tasks. After that, they started to walk through the park, with the participant walking just behind the experimenter. When a landmark became visible, the experimenter stopped for about 5 s, named the landmark and pointed at it (saying “There are the rushes on the left” in the SN-learning condition, or “There is the listening point on the right” in the NS-learning condition, for instance). Participants in the SN-learning condition followed the experimenter northwards from Gate 1 in the south to Gate 2 in the north. Participants in the NS-learning condition followed the experimenter southwards from Gate 2 in the north to Gate 1 in the south. The whole length of the path was covered only once, taking about 4-5 min (a time considered sufficient on the strength of a previous pilot study). The experimenter and participant then walked back to the lab at the psychology department, without speaking explicitly about the ongoing study (while they walked to and from the park). At the lab, the participant completed the pointing task first, then the map-drawing task (so that they could not see a map view of the park before completing the pointing task, as it could influence the orientation of their mental representation [12]). Then the sOPT and SDSR were completed in a balanced order across participants.

## 3. Results

The data analysis was conducted with the R software (Foundation for Statistical Computing: Vienna, Austria) [53]. Concerning the ability to point to the cardinal north correctly, more of the participants in the SN-learning condition succeeded in doing so (*N* = 34, as opposed to *N* = 12 who failed), while the proportions of participants in the NS-learning condition who succeeded in identifying the cardinal north was much the same (*N* = 23 and *N* = 22, respectively) (χ^2^
_(1)_ = 4.13, *p* = 0.042). The map-drawing task showed that accuracy (SQRTCO) did not differ by learning condition, i.e., it was similar for SN-learning (M = 0.88, SD = 0.17) and NS-learning (M = 0.93, SD = 0.08), F(1, 89) = 2.81, *p* = 0.09. Map-drawing performance, as an indicator of participants’ mental representations of a configuration, was very good in both groups, i.e., all participants positioned the landmarks on the sketch map perfectly or almost perfectly. It was consequently not considered in the further analyses to avoid the results suffering from the ceiling effect. Accuracy in the map-drawing task correlated negatively with pointing errors, however (r = −0.24, *p* < 0.05). As for the visuo-spatial factors, two preliminary ANOVAs ascertained that the two groups did not differ in terms of sOPT (F < 1, *p* = 0.41) and SDSR (F (1,89) = 1.58, *p* = 0.21) scores (see Table 1). Table 1 shows the means and standard deviations, and the correlations between all measures for the two groups.

### Pointing Task

Linear models were run for pointing errors. Gender (given its known influence on navigation performance [34,54]), and knowledge of the cardinal north (given the difference between our SN-learning and NS-learning groups) were entered as control variables in a baseline model (step 0). Then the path learning condition (SN vs. NS), the type of pointing (SN vs. NS), and the learning condition × type of pointing interactions were entered in the model (step 1). The sOPT and SDSR were then added in a final model (step 2). The AIC (Akaike Information Criterion [55]) was considered to indicate the models’ goodness of fit, with decreasing AIC values when the predictors were entered indicating improvements to the model. Table 2 shows the AIC, standardized coefficients, and p-values of the predictors for Steps 0, 1, and 2. The model in Step 0 showed that gender and knowledge of the cardinal north were not significant predictors. The model in Step 1 showed a significant role of: learning condition (β = 0.27, *p* = 0.009), SN-learning coinciding with fewer degrees of error than NS-learning; type of pointing (β = 0.34, *p* = 0.001), SN pointing coinciding with fewer degrees of error than NS pointing. The learning condition x type of pointing interaction was significant (β = −0.35, *p* = 0.006), indicating: a greater accuracy in SN pointing (fewer degrees of error) than in NS pointing in the SN-learning condition; and a similar performance for SN and NS pointing in the NS-learning condition. This interaction is illustrated in Figure 2. Further, the model in Step 2 showed the significant contribution of sOPT, after accounting for the predictors added in the previous step (β = −0.37, *p* < 0.001), which indicates that a greater ability to adopt imaginary views in the sOPT (fewer degrees of error) was associated with a better SN and NS pointing performance (fewer degrees of error). The contribution of SDSR was not significant. The role of the sOPT in relation to learning condition and type of pointing was also explored, but the interactions were not significant (*p_s_* > 0.08), so they were not included in the final model. The final model (step 2) explained 19% of the total variance.

## 4. Discussion

This study aimed to examine the formation of environment representations acquired by navigation in terms of: a) the interplay between egocentric experience and allocentric (world-based) information, i.e., whether the initial egocentric view or a combination of the egocentric with allocentric (world-based) information prevails; and b) the contribution of individual visuo-spatial factors. An important aspect of learning from navigation that has yet to receive much attention concerns the role of reference systems (egocentric vs. allocentric [world-based] frames of references) used to organize spatial information in memory, and the contribution of individual visuo-spatial factors in supporting environment representations.

A real environment was considered in an effort to shed light on these issues, comparing our participants’ recall of a path travelled so that the initial heading and main orientation (egocentric experience) was aligned with either the cardinal north (SN-learning condition) or the cardinal south (NS learning condition). In the SN-learning condition, there is an alignment between the initial view (personally north-up) and the compass north-oriented information. The NS-learning condition, on the other hand, creates a potentially misalignment between the initial view (personally north-up) and the compass south-oriented information. These two different learning conditions enabled us to investigate the interplay between egocentric (initial heading) and allocentric (world-based) frames of reference, and to see how they work in organizing spatial information in memory. The contribution of individual visuo-spatial factors in supporting mental representations of an environment was also examined.

Concerning the role of egocentric and allocentric (world-based) frames of reference, the results of our linear models showed a generally better performance: if the path was experienced heading north rather than south, i.e., the SN-learning group had fewer degrees of error in pointing than the NS-learning group; and for imagined northward heading than for imagined southward heading, with fewer degrees of error in SN pointing than in NS pointing. The effects of learning condition and type of pointing were better qualified by their interaction: when participants walked northwards along the path (SN-learning), their performance was better (fewer degrees of error) for SN pointing than for NS pointing, but when participants walked southwards (NS-learning), their performance in SN and NS pointing was similar. These findings are in line with the hypothesis according to which, when we learn new environments by navigating them, our mental representations incorporate both our initial view and allocentric information, in terms of cardinal directions. Ensuring an initial view aligned with an allocentric northward orientation (by explicitly telling participants they were heading north before they started to walk along the path) produces a representation with a clear cardinal north orientation. The alignment of the initial view (based on a personal north-up orientation) with the cardinal north prompted a strongly north-oriented representation. When the initial view was cardinal south oriented (by explicitly telling participants they were facing south before starting along the path) there was no detectable preferential orientation attributable to either the initial view or the allocentric information provided.

These results hold after accounting for knowledge of the cardinal north in the field and gender (which had no significant effects on pointing performance, however). Concerning knowledge of the cardinal north in the field, it should be noted that most of our participants were able to indicate the cardinal north in the environment considered, especially those in the SN group (who subsequently faced north as they walked along the path). This might be due to the appeal of facing north [21,22].

Overall, our results support the assumption that the initial view is important in the orientation of our mental representations of an environment [7,19,26], but allocentric information plays a part as well. To be more specific, world-based references in terms of a cardinal north orientation are important in boosting this orientation of environment representations in our memory. Our findings are consistent with Gagnon and colleagues’ results indicating that, if participants were shown which way was north (with an image of a compass rose) before they embarked on a virtual navigation, their representations had a preferred north-up orientation [28]. The same applied if participants were told which way was north in an imaginary tour [29,30,31]. This property was only assessed with a pointing task in the latter studies. These results deriving from a combination of navigation towards the real cardinal north (as a learning modality) and pointing (as an assessment modality) contribute to confirming the assumption that representations are north-up oriented. This interplay of initial view with cardinal north seen in our participants confirms the findings of previous studies [28,29,30], and also extends them to real-world environments with a real cardinal north. These results also support the assumption that cardinal north is appealing for orienting our representations of newly-acquired environments as well as familiar ones [21,22], even though the role of the cardinal north orientation in environment representations is still questioned [25].

Although these findings seem to clearly indicate an effect on the orientation of our participants’ representations when there was an alignment of their initial view with the cardinal north, the outcome when there was no such alignment, i.e., when their initial (personally north-up) view was not aligned with the cardinal north (i.e., south-oriented), was not so clear-cut. This mismatch would interfere with the emergence of the initial learning view [18,30], without revealing any clear role of the allocentric frame of reference. In short, mismatching egocentric and allocentric (world-based) information is an issue that deserves to be better examined in future studies. It should be noted that some features of the environment chosen might help to explain why no clear picture emerged of what happens with a south-oriented path. Given that the visibility and arrangement of landmarks influence our environment representations [e.g., 18,19], the fact that the perspective from which the landmarks were viewed changed according to the learning condition, and that they were not equidistant from one another (because the environment was not designed ad hoc) could affect the representation. This might explain, at least in part, why no clear results emerged when the environment was experienced according to a south-up orientation.

As for the role of individual visuo-spatial factors, our results mainly showed that the accuracy of mental representations of a newly-learnt environment (as assessed with aligned and counter-aligned pointing) was predicted by participants’ perspective-taking ability, i.e., their individual flexibility in adopting different imaginary views. Although perspective-taking ability supported mental representation accuracy in our sample (as previously suggested [41,42,43]), this effect was not more obvious when participants’ imaginary view in the test phase was misaligned with that of the initial learning phase [29,30]. Among several explanations, the path’s regularity and consistency with the park’s layout may have a role in reducing the involvement of individual ability, which may increase in the case of complex paths and layouts [29,30]. Their sense of direction was expected to affect their pointing performance [27,44], but its role proved to be marginal: the correlation went in the right direction but was not significant. This result is consistent with previous studies that considered both visuo-spatial abilities and self-reported spatial preferences [32,35], finding only a minor role for the latter. Self-perceived sense of direction is known to influence navigation [32], and although environmental features and learning condition might alter its influence, its role warrants further investigation.

While our results are interesting and offer new insight, they need to be circumscribed within the geographical context in which the study was carried out, i.e., in a European country, in a northern region of Italy. The path considered was learnt in a garden that has quite a regular layout, with no salient external landmarks that could serve as reference points, and provide contextual information that might prompt participants to adopt a north-up organization of their mental representations. There is evidence to suggest that representations are not north-up per se, however. Relevant geographical landmarks (such as rivers and mountains) can be used as global viewer-independent frames [27,56]. The geographical positions of different countries and the language used [57,58,59] also influence the features of environment representations. For instance, Madaleno [57] analyzed environment representations in people living in countries in the southern hemisphere, and showed that some of them formed south-up mental maps. The use of a language based on a geocentric frame (the equivalent of geographical directions), typical of certain populations living in countries with a particular topography (such as islands), and characteristic landmarks (such as mountains or volcanoes), also shapes the proprieties of people’s environment representations (which are not necessarily cardinally north-up oriented; [58,59]).

Future studies should expand our results to different cultural and geographical backgrounds. Others, within the same geographical setting, will be needed to corroborate the role of egocentric and allocentric information in more depth by manipulating the features of a path in relation to those of the surrounding environment. While our path was fairly straight and aligned with the surrounding layout, it would be useful to see whether the results change for more complex paths (with more changes of direction [12]), or when the path’s shape is dissonant with the layout of the environment [18]. This would help to clarify whether the role of the cardinal north remains important, or whether other environmental features to be integrated in our mental representations become more relevant [25,27]. Another possibility would be to include the experience of other headings (such as east and west [30]), or both north and south headings to shed more light on the role of initial headings vis-à-vis environmental features in environment representations. The same park setting chosen for our study would be appropriate for testing at least some of these options by getting participants to start walking northwards, for instance, and then change course and walk towards the east or west, or turn back and head south. Further research also needs to examine the effect of orientation, and its flexibility, using not only pointing tasks, but also participants’ ability to make spatial inferences and freely explore an environment [28]), and to draw maps. The map drawing task used in the present study served as a measure of accuracy but did not provide information about the orientation of participants’ mental maps. This probably occurred because of the environment’s features (such as its regularity and the limited number of landmarks), and our method (giving participants a sketch map to place the landmarks on that was already oriented as in the learning condition). A map drawing task should be used in future studies with a more complex environment and ask participants to draw their own map (instead of giving them a layout to place landmarks on) in order to capture its organization properties. Finally, given the influence of individual differences on learning from navigation [33,34,35,36], further studies should seek to clarify how other individual factors, such as gender (given the larger number of females in this study) and other visuo-spatial abilities and attitudes, such as stance on wayfinding [48], might contribute to explaining the properties of our environment representations. Given that the variance explained by our model was only moderate, we need to clarify which factors (both environmental and individual) can best capture the features of an environment representation formed from navigation.

## 5. Conclusions

To conclude, this report contributes to broadening the spatial cognition framework to include the factors favoring environment representations acquired by navigation. The results mainly showed that: (i) learning a path with an initial heading aligned with the cardinal north promotes a clearly north-oriented mental representation of the environment; and (ii) this representation is supported by individual perspective-taking ability. These are, among others, some of the key factors that contribute to optimizing environment learning from navigation.

## Figures and Tables

**Figure 1 brainsci-10-00204-f001:**
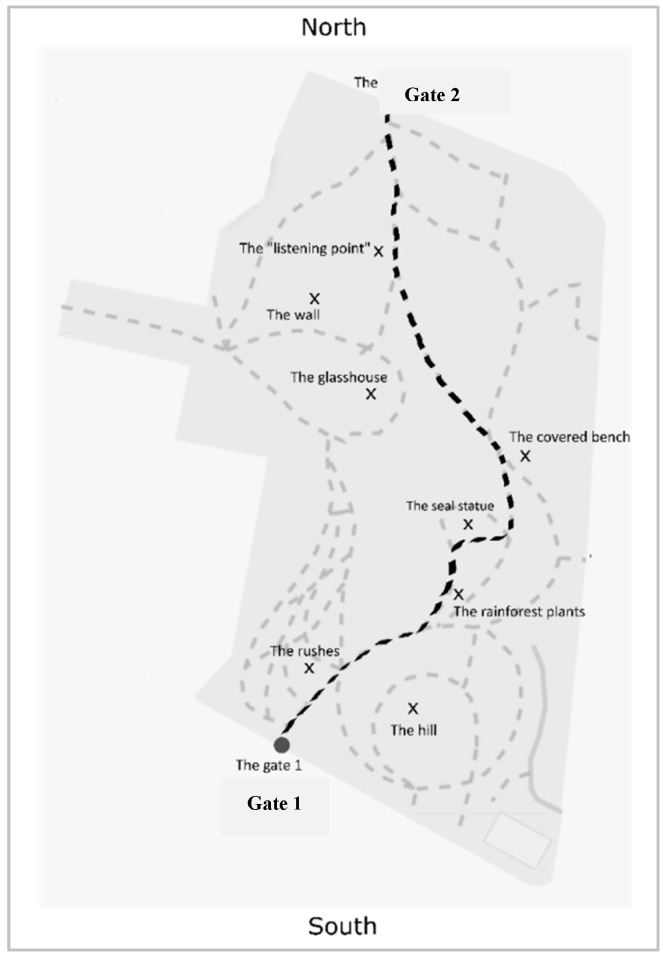
Map of the Europe Park (Padova, Italy), showing the path (black dashed line) and the eight landmarks considered (marked with an X), Gate 1 (to the south) and Gate 2 (to the north). Map retrieved and adapted from www.openstreetmap.org.

**Figure 2 brainsci-10-00204-f002:**
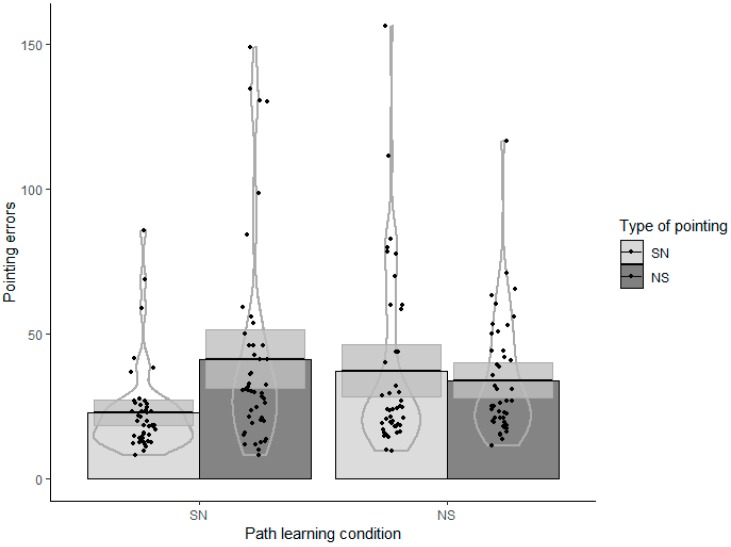
Pointing task. Learning condition × Type of pointing (SN = south to north; NS = north to south).

**Table 1 brainsci-10-00204-t001:** Means and standard deviations by learning condition (SN = south to north or NS = north to south) and correlations between all measures of interest.

	SN-Learning		NS-Learning		1	2	3
	*M*	*SD*	*M*	*SD*			
1. sOPT (max. 180° ^a^)	45.42	26.36	41.39	19.73	-		
2. SDSR (max. 65)	44.33	7.96	46.40	7.77	−0.183	-	
3. SN pointing (max. 180° ^a^)	22.74	15.07	37.21	30.65	0.313 **	−0.038	-
4. NS pointing (max. 180° ^a^)	41.27	34.68	33.93	20.17	0.368 ***	−0.107	0.130

sOPT: Short Object Perspective Taking; SDSR: Sense of Direction and Spatial Representation. ^a^ Degrees of error. ** *p* ≤ 0.01; *** *p* ≤ 0.01.

**Table 2 brainsci-10-00204-t002:** Pointing task. Steps in the linear regression models with AIC (Akaike Information Criterion), predictors β and p-values.

	AIC	β	*p*
Step 0	1723		
Gender		0.03	0.646
Knowledge of the cardinal north		−0.00	0.988
Step 1	1717		
Learning condition		0.27	0.009
Type of pointing		0.34	0.001
Learning condition × type of pointing		−0.35	0.006
Step 2	1695		
sOPT		0.37	<0.001
SDSR		−0.01	0.889

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
