# Peer review of "Learning a Path from Real Navigation: The Advantage of Initial View, Cardinal North and Visuo-Spatial Ability"

_brainsci, 2020, doi:10.3390/brainsci10040204_

Round 1
Reviewer 1 Report
Review of MS brainsci-733530 by Muffato and Meneghetti – Learning a path from real navigation…
This is a fine study. The Introduction by itself could be a sort of review or mini-review on orienting and navigating, and the authors should be commended for that. However, they may wish to consider the following comments.
- Hasn't the study of Ganon et al (L.132 and on) already described the main finding of the present study? Please make clear what the novelty is in the present study and what will be a follow-up or reconfirmation of past studies/ideas. This is especially important considering the relevant Discussion section (L.414-425), indicating that this study is more of a support for past findings/ideas.
- Could it be that the difference in the order of landmarks and their perspective are the source of the difference between the SN and NS groups? In other words – is the cardinal north really isolated from other factors in this study? For such a conclusion the landmarks need to be identically spaced and viewed for both SN and NS groups. This is less likely to affect the pointing and sense of direction but such effect was not ruled out (all these are surmised in L.433-466).
- Was testing based on individual participants, or testing a group (2 or more) each time)? Please make this clear (written in plural voice, it is as if this was a group). If this was a group, then there could have been a strong social effect (depending on the interactions among participants). In the same vein, what was the mode of interaction between the experimenter and participant during the test? No talking except for the spatial info?
- The aims are introduced in a very general way (e.g. – L. 181-185 or 187-190) – they must be stated more explicitly, even if there is a section (1.3). Specific questions that were asked in this test should be posed. In the context, section 1.3 provides information that belongs to the 'Methods' section and for brevity should be removed from here. This information has no value anyway without the details of the method. For example, in section 1.3 one may wonder whether the test environment was unfamiliar to the participants (L. 196). This is clarified in the only later in the 'Methods'
- The introductory paragraph of the 'Introduction' is first L.34 does not include references – for Example. L.34 makes a statement that should refer to supporting evidence (e.g. – references 17-20, 24-26 in this study). Alternatively, mention that this is discussed in detail below.
- Perhaps references to animals that use various forms of the compass could set a hypothesis that humans may possess an alike capacity.
- The authors may wish to explain that the egocentric frame of reference is polar, whereas the global (or allocentric) one is basically Cartesian. A north compass could be an integrator of such two relatively independent systems.
- 117-120 – this is the target of the article; introducing and emphasizing it earlier would clarify the target of the study from the beginning.
- 241 – is there a possible gender bias? How were the genders divided in each test group?
- 305 – did the participants know the aim of the test? Were they instructed to memorize the information provided by the experimenter when traveling in the park?
Finally, I wonder if there is a possible "cultural" effect that could affect the present results, considering that all maps and navigation means are aligned for reading north-wise.
Author Response
Dear Reviewer 1,
Below we reported the answers to all your comments. In “Report Notes” you can find the revised manuscript attached. All changes are marked in red.
This is a fine study. The Introduction by itself could be a sort of review or mini-review on orienting and navigating, and the authors should be commended for that. However, they may wish to consider the following comments.
Hasn't the study of Ganon et al (L.132 and on) already described the main finding of the present study? Please make clear what the novelty is in the present study and what will be a follow-up or reconfirmation of past studies/ideas. This is especially important considering the relevant Discussion section (L.414-425), indicating that this study is more of a support for past findings/ideas.
Answer: Thanks to your observation, we have better described the Gagnon et al. study in the Introduction, anticipating some of the aspects in which ours differs (such as establishing the cardinal orientation by displaying the compass rose at the beginning of the navigation phase, and not using a pointing task) (see pages 3 and 4). The novelty of our findings (regarding navigation in combination with the use of pointing) has also been better explained in the Discussion (see page 11).
Could it be that the difference in the order of landmarks and their perspective are the source of the difference between the SN and NS groups? In other words – is the cardinal north really isolated from other factors in this study? For such a conclusion the landmarks need to be identically spaced and viewed for both SN and NS groups. This is less likely to affect the pointing and sense of direction but such effect was not ruled out (all these are surmised in L.433-466).
Answer: In the light of this observation, it is important to mention that we used a real environment with already existing landmarks, which were not equidistant from one another. All landmarks were visible along the path, however, though - as you mentioned - the perspective and spacing was different when the path was walked from north to south or vice versa. What we know is that, in both conditions, the landmarks were well recalled and appropriately positioned (≥ 88%; see results section). That said, we cannot say for sure that the non-standard layout of the landmarks and the perspective view did not affect the orientation of the representations. This might explain, at least in part, why no clear results emerged when the environment was experienced heading south, as now mentioned in the Discussion (see page 11-12). Concerning the role of sense of direction, we had provided an explanation for the limited effect of sense of direction on page 12: “This result is consistent with previous studies that considered both visuo-spatial abilities and self-reported spatial preferences [34,35], finding only a minor role for the latter.”. We have now added that whether environmental features and learning condition might alter its effect is an issue that should be further investigated (see page 12).
Was testing based on individual participants, or testing a group (2 or more) each time)? Please make this clear (written in plural voice, it is as if this was a group). If this was a group, then there could have been a strong social effect (depending on the interactions among participants). In the same vein, what was the mode of interaction between the experimenter and participant during the test? No talking except for the spatial info?
Answer: All participants were tested individually, both in the field session and in the lab (as mentioned in the previous version, in the first sentence regarding the procedure). To avoid any risk of confusion, we have now reworded this sentence. The only interaction was with the Experimenter and participants were explicitly asked not to speak about the study while they went from department to park and back again (See page 8).
The aims are introduced in a very general way (e.g. – L. 181-185 or 187-190) – they must be stated more explicitly, even if there is a section (1.3). Specific questions that were asked in this test should be posed. In the context, section 1.3 provides information that belongs to the 'Methods' section and for brevity should be removed from here. This information has no value anyway without the details of the method. For example, in section 1.3 one may wonder whether the test environment was unfamiliar to the participants (L. 196). This is clarified in the only later in the 'Methods'
Answer: Thank you for your observation. We have better defined the aims of the study at the end of section 1.1 (see page 4) and at the end of section 1.2. (see page 5), at the beginning of the paragraph on the rationale of the study (see page 5), and in the Discussion (see page 10).
In section 1.3 we have moved part of the description of the study under “Procedure”, but left some information (about the path learning condition, i.e. south to north vs north to south, and the pointing used) because it makes it easier to understand the expected results presented in the hypotheses. We had already mentioned that the environment was unfamiliar to participants “A group of undergraduates walked for the first time along the same path through the park”. To avoid any risk of confusion, the fact that the path was unfamiliar has not been introduced earlier (see page 5).
The introductory paragraph of the 'Introduction' is first L.34 does not include references – for Example. L.34 makes a statement that should refer to supporting evidence (e.g. – references 17-20, 24-26 in this study). Alternatively, mention that this is discussed in detail below.
Answer: This was a very introductory paragraph that we wrote to offer an overview of the paper. We have now added some general references concerning the role of environmental and individual factors in forming a representation (see page 1).
Perhaps references to animals that use various forms of the compass could set a hypothesis that humans may possess an alike capacity.
Answer: We have added some references on the ability of animals to use egocentric and allocentric information to orient themselves (see page 2).
The authors may wish to explain that the egocentric frame of reference is polar, whereas the global (or allocentric) one is basically Cartesian. A north compass could be an integrator of such two relatively independent systems.
Answer: Thank you for your comment. When we introduce the egocentric frame and allocentric frames we now specify that they can be intended as a Polar reference system and Cartesian reference system, respectively (see page 2).
117-120 – this is the target of the article; introducing and emphasizing it earlier would clarify the target of the study from the beginning.
Answer: The core aspect of this research was also mentioned in the introductory paragraph (see page 1)
241 – is there a possible gender bias? How were the genders divided in each test group?
Answer: When gender was input as a control variable the linear model did not show a significant effect, meaning that gender did not affect pointing performance. Males and females were distributed so as to make sure that they were equally represented in both learning conditions (e.g. the first female was assigned to the SN learning condition, the second to the NS learning condition, and so on, and the same was done with the males). This has been explained in the procedure (see page 8). The role of gender as an individual difference should be better considered in further studies, however (as now mentioned on page 13).
305 – did the participants know the aim of the test? Were they instructed to memorize the information provided by the experimenter when traveling in the park?
Answer: We have now stated clearly in the procedure that the consent form signed by participants explained the aim of the study and the procedure involved, and that the experimenter instructed them to memorize the path they walked along and the elements named and pointed out (see page 8).
Finally, I wonder if there is a possible "cultural" effect that could affect the present results, considering that all maps and navigation means are aligned for reading north-wise.
Answer: You raise a relevant question. The fact that maps and navigation devices are oriented as north up can create a general bias in prompting representations based on a north-up organization. Then there are cultural and geographical factors that can intervene in reducing this north-up organization.
In the Discussion (see page 12, we have set our results in the geographical context in which the study was carried out, i.e. in a European country, in a northern region of Italy. The path considered was learnt in a garden that has quite a regular layout, with no salient external landmarks that could serve as reference points, and provide contextual information that might prompt participants to adopt a north-up organization of their mental representations. There is evidence to suggest that representations are not north-up per se, however. Relevant geographical landmarks (such as rivers and mountains) can be used as global viewer-independent frames (Brunyé et al., 2015; Meakins et al., 2016). The geographical positions of different countries and the language used (Madaleno, 2010; Palmer, 2002; Dansen et al., 2010) also influence the features of environment representations. For instance, Madaleno (2010) analyzed environment representations in people living in countries in the southern hemisphere, and showed that some of them formed south-up mental maps. The use of a language based on a geocentric frame (the equivalent of geographical directions), typical of certain populations living in countries with a particular topography (such as islands), and characteristic landmarks (such as mountains or volcanoes), also shapes the proprieties of people’s environment representations (which are not necessarily cardinally north-up oriented (Palmer, 2002; Dansen et al., 2010). We have therefore added that our results may depend to some degree on the geographical area considered.
Reviewer 2 Report
This study was nicely set up, the rationale was laid out, and methods and results were clearly presented. A few comments: the Methods should state what instructions or orienting comments were given to participants. That is, were they told what to focus on or how they would be tested later? Just a walk in the park? Any guidance that defined their task?
How was the map for recall presented to the Participants? was it given to everyone in a North up position? Did the NS participants turn it around?
In Results, you focus on developing a model for pointing, but the map recall task is interesting as well. While the primary experimental manipulation (SN/NS) did not predict differences, it would be interesting to know what DID predict differences in map recall. and, how map recall and pointing were related.
I was surprised that you are only accounting for 19% of the variance with what seem like a good set of predictors. What do you think is missing? or is it just a difficult task that is subject to many individual differences?
Minor notes: line 361, i think you meant to say Figure 2, not Fig 1; line 391 uses a colon that does not seem to belong.
Author Response
Dear Reviewer 2,
Below we reported the answers to all your comments. In “Report Notes” you can find the revised manuscript attached. All changes are marked in red.
This study was nicely set up, the rationale was laid out, and methods and results were clearly presented.
A few comments: the Methods should state what instructions or orienting comments were given to participants. That is, were they told what to focus on or how they would be tested later? Just a walk in the park? Any guidance that defined their task?
Answer: Thank you for your comment. In the procedure we have better explained that the experimenter instructed participants to memorize the path they would walk along, and all the elements being named and pointed out, as this information would be used in a task afterwards. We have also specified that information about the aim of the study and the procedure was presented in the consent form (see page 8).
How was the map for recall presented to the Participants? was it given to everyone in a North up position? Did the NS participants turn it around?
Answer: Thank you for this comment, which prompted us to clarify this aspect. The sketch map was presented following the same orientation as the learning condition (as now mentioned in the presentation of the sketch map, page 7), i.e., participants in the SN condition were presented with a north-up sketch, while participants in the NS condition were presented with the same map rotated so that it was south-up.
In Results, you focus on developing a model for pointing, but the map recall task is interesting as well. While the primary experimental manipulation (SN/NS) did not predict differences, it would be interesting to know what DID predict differences in map recall. and, how map recall and pointing were related.
Answer: Thank you for your comment. Accuracy in the map-drawing task is interesting as well, we agree, but there was no difference in this variable between the SN and NS learning conditions. The SQRTCO was similar for SN-learning (M= 0.88, SD = 0.17) and NS-learning (M = 0.93, SD = 0.08), F(1, 89) = 2.81, p = 0.09), meaning no effect of learning condition. In other words, participants’ mental representations of the configuration were very good in both cases. To avoid our results suffering from the ceiling effect, we opted not to consider the map drawing task in our further analyses. This information has been reworded (see page 9). As suggested, we have added the correlation between accuracy in the map-drawing task and pointing errors (r= -0.24, p < 0.05). This information has been added (see page 9).
Although the map drawing task was not the core task in our study, and was only administered to ascertain whether people could locate the landmarks accurately on a map - i.e., as a control task to avoid the risk of participants failing to recall the landmarks - we agree that map drawing can be a very interesting measure for understanding how a mental map is oriented. Our providing information relating to the environment (such as its regularity and the limited number of landmarks), and our method (giving participants a sketch to place the landmarks on) probably prevented us from inferring anything about the orientation of participants’ mental maps. A map drawing task should be used in future studies with a more complex environment and asking participants to draw their own map (not using any layout as reference) in order to capture its organization properties. This is now reported in the discussion (see page 13).
I was surprised that you are only accounting for 19% of the variance with what seem like a good set of predictors. What do you think is missing? or is it just a difficult task that is subject to many individual differences?
Answer: We agree that the variance explained is only moderate, though other studies on navigation found a similar share of variance explained (values ranging 10-20%, Muffato & Meneghetti, 2020; Meneghetti et al., 2017). Certainly, there are other factors not considered in the present study, such as gender (in our study, the two sexes were not equally represented, as Reviewer 1 also mentioned), and other visuo-spatial abilities and attitudes relating to environmental features, that need to be investigated to better capture people’s environment representations. This is now presented in the Discussion (see page 13).
Minor notes: line 361, i think you meant to say Figure 2, not Fig 1; line 391 uses a colon that does not seem to belong.
Answer: Thank you. Both errors have been corrected.